# Factors influencing influenza, pneumococcal and shingles vaccine uptake and refusal in older adults: a population-based cross-sectional study in England

Pui San Tan [ID],[1] Martina Patone [ID],[1] Ashley Kieran Clift [ID],[1] Hajira Dambha-Miller,[2] Defne Saatci,[1] Tom A Ranger [ID],[1] Cesar Garriga [ID],[1] Francesco Zaccardi,[3,4] Baiju R Shah,[5] Carol Coupland,[6] Simon J Griffin,[7] Kamlesh Khunti,[4] Julia Hippisley-Cox [ID] [1]

PST, MP and AKC contributed equally.

For numbered affiliations see end of article.

**Correspondence to**
Dr Pui San Tan;
pui.tan@phc.ox.ac.uk

## ABSTRACT

**Objectives** Uptake of influenza, pneumococcal and shingles vaccines in older adults vary across regions and socioeconomic backgrounds. In this study, we study the coverage and factors associated with vaccination uptake, as well as refusal in the unvaccinated population and their associations with ethnicity, deprivation, household size and health conditions.

**Design, setting and participants** This is a cross-sectional study of adults aged 65 years or older in England, using a large primary care database. Associations of vaccine uptake and refusal in the unvaccinated with ethnicity, deprivation, household size and health conditions were modelled using multivariable logistic regression.

**Outcome measure** Influenza, pneumococcal and shingles vaccine uptake and refusal (in the unvaccinated).

**Results** This study included 2 054 463 patients from 1318 general practices. 1 711 465 (83.3%) received at least one influenza vaccine, 1 391 228 (67.7%) pneumococcal vaccine and 690 783 (53.4%) shingles vaccine. Compared with White ethnicity, influenza vaccine uptake was lower in Chinese (OR 0.49; 95% CI 0.45 to 0.53), 'Other ethnic' groups (0.63; 95% CI 0.60 to 0.65), black Caribbean (0.68; 95% CI 0.64 to 0.71) and black African (0.72; 95% CI 0.68 to 0.77). There was generally lower vaccination uptake among more deprived individuals, people living in larger household sizes (three or more persons) and those with fewer health conditions. Among those who were unvaccinated, higher odds of refusal were associated with the black Caribbean ethnic group and marginally with increased deprivation, but not associated with higher refusal in those living in large households or those with lesser health conditions.

**Conclusion** Certain ethnic minority groups, deprived populations, large households and 'healthier' individuals were less likely to receive a vaccine, although higher refusal was only associated with ethnicity and deprivation but not larger households nor healthier individuals. Understanding these may inform tailored public health messaging to different communities for equitable implementation of vaccination programmes.

## STRENGTHS AND LIMITATIONS OF THIS STUDY

⇒ Use of a large primary care database offered a population-representative population in terms of demographics including ethnic groups and deprivation.

⇒ Using a primary care database captured comprehensive vaccination data, including those occurring outside general practice (such as in pharmacies), as well as recorded invitations to vaccination sent by general practices and patient refusals.

⇒ There was lack of recording of variables such as personal beliefs, literacy levels, language barriers, access and education status, and hence we were unable to evaluate the impact of these factors on vaccination uptake and refusal.

⇒ Classification of vaccination-related endpoints was reliant on individual practitioners; however, we used an appropriately wide range of codes in our endpoint definitions.

## BACKGROUND

Older adults are often more susceptible to infectious diseases circulating in the community, and may develop more severe health outcomes when infected due to lower immune responses associated with ageing[1] and comorbidities. National influenza, pneumococcal and shingles vaccination programmes for older adults have been implemented in the UK in various phases.[2–4] Through these national vaccination programmes, 'seasonal' influenza vaccines are offered annually, pneumococcal vaccines are offered as a single dose to adults aged 65 years and above, while the shingles vaccine is offered as a single dose to adults aged 70–79 years.[2–4]

The WHO recommends a target of 75% population vaccination coverage.[5] Recent reports from Public Health England have

reported 81% influenza vaccination coverage and 69% pneumococcal vaccination coverage in adults aged 65 years and above, and 47%–77% for shingles vaccination coverage in adults aged 71 and 78, respectively.[2–4] However, some evidence suggests that there could be differences in terms of vaccination coverage, potentially varying by geographical region, ethnicity, deprivation, household size and health conditions.[2–4 6 7]

For the purposes of equitable public health strategy, it is important to understand factors associated with uptake of vaccinations, and refusal of vaccinations in the unvaccinated population. Prior studies have demonstrated differential uptake of existing vaccinations across sociodemographic groups, however, many studies have either studied single vaccinations, not captured the appreciable casemix inherent to sociodemographic groups (such as by using broad ethnic categories), analysed a small set of relevant health conditions, and relied on potentially imprecise or biased self-report measures.[7–9] In addition, although household size is known to increase the risk of transmission for infectious diseases, evidence on the association between household size and vaccination uptake remains limited.[10] A few previous studies have suggested that individuals from larger households were less likely to be vaccinated, although these studies were small and mainly focused on childhood vaccinations.[11 12] Further, it is of interest to understand the pathway events leading to the lack of vaccine uptake, and to what extent these are driven by patient refusal.

Here, we evaluated factors associated with uptake and refusal of existing national vaccination programmes (influenza, pneumococcal and shingles) in older adults (aged 65 years and above) in England and their associations with ethnic group, deprivation, household size and health conditions.

## METHODS
### Study population and data source
We performed a population-based cross-sectional study using QResearch (V.45). QResearch is a database with over 10 million current patients registered at more than 1800 practices in England. QResearch is an electronic healthcare primary care database in the UK with individual patient level records for general practices using the EMIS computer record system. The database captures information from general pratitioner (GP) consultations; including patient demographics, socioeconomic status, diagnoses, laboratory test results, treatments and vaccinations. The database has good representation of the general population of England, particularly in terms of different ethnic groups with proportions close to those reported by Office for National Statistics.[13]

In this study, we included adults aged 65–99 years currently registered with 1318 practices during the period 24 January 2020 to 31 October 2020, which comprised 2 054 463 of approximately 13.7 million patients aged 65 and over registered with a GP in England.[14] We assessed the uptake and refusal of influenza, pneumococcal and shingles vaccines from 1 January 1989 to 31 October 2020 (last database update) as our main study outcome. As the shingles vaccination was rolled out nationally in England in 2013 for those aged 70 and up until 79,[15] we included in our shingles vaccine analysis only those aged 70 and above, excluding those aged 80 and above in year 2013 as they were not eligible at the time. Uptake was defined as the last recorded instance of receiving the vaccines of interest within the study period. This was mostly in GP surgeries (~99%), but also in-hospital or pharmacy administrations. Refusal was analysed in those with no record of vaccination, defined as last recorded instances of explicit refusal (74%–82% of recorded code instances), consent not being given (18%–26%) or non-attendance to a scheduled vaccination appointment (0.03–0.3%).[16] Outcomes were defined using code dictionaries comprising relevant Read and SNOMED codes as inputted into the EMIS software by healthcare practitioners.

We extracted demographic data including age, sex, self-reported ethnic group, Townsend deprivation index quintile,[17 18] geographical region within England (n=10, see table 1), housing status and household size. Townsend deprivation score is commonly used in the UK to measure socioeconomic status. It uses the following characteristics to measure deprivation by postcode; proportion of (1) unemployment, (2) non-car ownership, (3) non-home ownership and (4) household crowding— a higher score suggesting greater deprivation. In this study, the scores were reported in quintiles, that is, the first quintile indicates the least deprived, while the fifth quintile indicates most deprived. Ethnicity was grouped into nine categories—white (white British, white Irish, other white), Indian, Pakistani, Bangladeshi, Other Asian, black Caribbean, black African, Chinese, Other ethnic group (white and black, white and Asian, other mixed, other black, other ethnic group). We also extracted data using GP Read and SNOMED codes from primary care records and International Classification of Diseases 10th Revision (ICD-10) codes from hospital records (where available) for diagnoses of asthma, chronic obstructive pulmonary disease (COPD), diabetes mellitus (types 1 and 2), hypertension, coronary heart disease, atrial fibrillation (AF), congenital heart disease, congestive cardiac failure (CCF), chronic neurological diseases (Parkinson's disease, epilepsy, cerebral palsy), learning disability, dementia and severe mental illness (schizophrenia, severe depression, bipolar affective disorder and psychosis) and immune suppression (based on use of immunosuppressant medications). For each vaccination outcome (uptake and refusal), people with health conditions diagnosed prior to the vaccination outcome were defined as exposed, while those diagnosed with health conditions after the outcome were defined as unexposed. The most recently recorded body mass index (BMI) and smoking status were identified for each individual.

**Table 1** Characteristics of study population in patients aged 65+ (70+ for shingles)

| Characteristics | | Study population | Vaccine uptake | | |
|---|---|---|---|---|---|
| | | Overall | Influenza | Pneumococcal | Shingles* |
| Total | N (row %) | 2 054 463 | 1 711 465 (83.3) | 1 391 228 (67.7) | 690 783 (53.4) |
| Age | Mean (SD) | 75.5 (7.7) | 76.3 (7.7) | 77.1 (7.5) | 77.2 (4.4) |
| | 65–69 | 541 272 (26.3) | 373 566 (21.8) | 232 831 (16.7) | – |
| | 70–79 | 922 198 (44.9) | 793 150 (46.3) | 665 037 (47.8) | 469 684 (68.0) |
| | 80–89 | 471 167 (22.9) | 434 074 (25.4) | 395 456 (28.4) | 221 099 (32.0) |
| | 90–99 | 119 826 (5.8) | 110 675 (6.5) | 97 904 (7.0) | – |
| Sex | Female | 1 100 957 (53.6) | 926 592 (54.1) | 749 022 (53.8) | 365 203 (52.9) |
| | Male | 953 506 (46.4) | 784 873 (45.9) | 642 206 (46.2) | 325 580 (47.1) |
| Ethnicity | White | 1 522 868 (74.1) | 1 293 856 (75.6) | 1 064 331 (76.5) | 539 237 (78.1) |
| | Indian | 35 618 (1.7) | 31 062 (1.8) | 25 454 (1.8) | 11 293 (1.6) |
| | Pakistani | 17 555 (0.9) | 15 588 (0.9) | 12 090 (0.9) | 4388 (0.6) |
| | Bangladeshi | 8138 (0.4) | 7635 (0.4) | 6264 (0.5) | 2076 (0.3) |
| | Other Asian | 17 848 (0.9) | 15 171 (0.9) | 11 890 (0.9) | 5135 (0.7) |
| | Black Caribbean | 22 859 (1.1) | 18 010 (1.1) | 14 102 (1.0) | 5791 (0.8) |
| | Black African | 16 880 (0.8) | 13 530 (0.8) | 9545 (0.7) | 3518 (0.5) |
| | Chinese | 6553 (0.3) | 4835 (0.3) | 3507 (0.3) | 1502 (0.2) |
| | Other ethnic groups | 25 410 (1.2) | 19 778 (1.2) | 14 569 (1.0) | 5832 (0.8) |
| | Ethnicity not recorded | 380 734 (18.5) | 292 000 (17.1) | 229 476 (16.5) | 112 011 (16.2) |
| Region | East Midlands | 46 002 (2.2) | 38 777 (2.3) | 30 526 (2.2) | 16 779 (2.4) |
| | East of England | 93 217 (4.5) | 77 645 (4.5) | 64 843 (4.7) | 34 167 (4.9) |
| | London | 322 941 (15.7) | 261 176 (15.3) | 204 112 (14.7) | 92 174 (13.3) |
| | North East | 47 496 (2.3) | 40 081 (2.3) | 33 271 (2.4) | 15 848 (2.3) |
| | North West | 417 970 (20.3) | 354 779 (20.7) | 292 600 (21.0) | 140 099 (20.3) |
| | South Central | 283 054 (13.8) | 239 109 (14.0) | 199 347 (14.3) | 102 632 (14.9) |
| | South East | 268 594 (13.1) | 220 952 (12.9) | 179 031 (12.9) | 91 516 (13.2) |
| | South West | 256 384 (12.5) | 213 037 (12.4) | 169 824 (12.2) | 87 179 (12.6) |
| | West Midlands | 237 881 (11.6) | 197 414 (11.5) | 161 606 (11.6) | 81 942 (11.9) |
| | Yorkshire & Humber | 80 924 (3.9) | 68 495 (4.0) | 56 068 (4.0) | 28 447 (4.1) |
| Deprivation quintile | 1 (most affluent) | 674 004 (32.8) | 569 701 (33.3) | 471 575 (33.9) | 251 660 (36.4) |
| | 2 | 547 862 (26.7) | 456 956 (26.7) | 373 336 (26.8) | 191 172 (27.7) |
| | 3 | 385 476 (18.8) | 318 962 (18.6) | 258 842 (18.6) | 123 090 (17.8) |
| | 4 | 267 458 (13.0) | 219 941 (12.9) | 175 665 (12.6) | 78 550 (11.4) |
| | 5 (most deprived) | 174 280 (8.5) | 141 551 (8.3) | 108 526 (7.8) | 44 651 (6.5) |
| | Not recorded | 5383 (0.3) | 4354 (0.3) | 3284 (0.2) | 1660 (0.2) |
| Home category | Neither in care home nor homeless | 2 005 725 (97.6) | 1 665 389 (97.3) | 1 356 313 (97.5) | 682 316 (98.8) |
| | Care home | 47 655 (2.3) | 45 263 (2.6) | 34 352 (2.5) | 8301 (1.2) |
| | Homeless | 1083 (0.1) | 813 (<0.01) | 563 (<0.01) | 166 (<0.01) |
| Household size | 1 person | 875 588 (42.6) | 726 447 (42.4) | 596 361 (42.9) | 285 715 (41.4) |
| | 2 people | 849 357 (41.3) | 721 411 (42.2) | 594 481 (42.7) | 326 499 (47.3) |
| | 3–5 people | 255 089 (12.4) | 199 611 (11.7) | 152 373 (11.0) | 65 031 (9.4) |
| | 6–9 people | 30 961 (1.5) | 24 934 (1.5) | 18 767 (1.3) | 6678 (1.0) |
| | 10 or more | 43 468 (2.1) | 39 062 (2.3) | 29 246 (2.1) | 6860 (1.0) |

**Table 1** Continued

| Characteristics | | Study population | Vaccine uptake | | |
|---|---|---|---|---|---|
| | | Overall | Influenza | Pneumococcal | Shingles* |
| No of health conditions† | 0 | 667 163 (32.5) | 483 507 (28.3) | 566 398 (40.7) | 213 919 (31.0) |
| | 1 | 786 798 (38.3) | 671 330 (39.2) | 559 648 (40.2) | 281 353 (40.7) |
| | 2 | 428 751 (20.9) | 393 220 (23.0) | 215 126 (15.5) | 145 583 (21.1) |
| | 3+ | 171 751 (8.4) | 163 408 (9.5) | 50 056 (3.6) | 49 928 (7.2) |

*Percentage calculated using denominator of shingles eligible population, n=1 294 176. Percentages are column percentages unless otherwise indicated.
†Counts only based on conditions included in this study.

## Analyses

Descriptive analyses compared the uptake and refusal of the three vaccinations of interest by ethnic group, Townsend deprivation quintiles, household size and individual health conditions. Percentage uptake of each vaccination in individual GPs was plotted to display between-region variations.

Multivariable logistic regression models examined associations between ethnic group, deprivation, household size, health conditions and vaccination uptake and refusal by calculating adjusted OR and their 95% CIs. Clustered robust SEs were used to account for clustering of individuals within GPs. Refusals were evaluated in never-receivers of each vaccine (no uptake). Individual models for each exposure (ethnic group, deprivation, household size, health conditions) and outcome (vaccination uptake and refusal for each vaccine) were fitted separately, allowing for adjustment of confounders: age, sex, geographical region, type of home, smoking status and/or BMI as relevant according to directed acyclic graphs—(1) ethnicity—no adjustments; (2) deprivation—adjusted for age, sex, region, ethnicity, household size; (3) household size—adjusted for age, sex, region, ethnicity, deprivation, (4) health conditions—age, sex, region, ethnicity, deprivation, household size, house type, smoking and BMI (online supplemental figure S1).

Missing data for ethnic group (18.5%), BMI (5.6%), deprivation quintiles (0.3%) and smoking status (1.0%) were multiply imputed using chained equations under the missing at random assumption. Five imputations were generated using a single rich imputation model incorporating all outcomes, exposures and confounder covariates. Models were fitted in each of the five imputed datasets with model coefficients and their SEs pooled in accordance with Rubin's rules.[19] We also performed sensitivity analyses of results using complete-case analysis.

In addition, we performed post-hoc interaction analyses to explore potential interactive effects for vaccine uptake between ethnicity and deprivation, household size and number of health conditions.

The reporting of studies using observational routinely-collected data (RECORD) guidelines were used for reporting.[20] Statistical analyses were performed using STATA V.17.0.[21]

## Patient and public involvement reporting

Two public representatives advised on interest and appropriateness of the research questions, were involved in writing the protocol for the wider study and input on lay-summaries describing the planned study.

## RESULTS

This study included 2 054 463 patients aged 65 years and older registered with 1318 GPs. Characteristics of the study population are shown in table 1 and S1. At least one influenza vaccine was received by 1 711 465 (83.3%) patients, a pneumococcal vaccine by 1 391 228 (67.7%) and a shingles vaccine by 690 783 (53.4% of over 70s). Figure 1 shows a descriptive overview of the rate of vaccination uptake and refusals by different regions in England at the practice level. For example, the median level of shingles vaccine uptake in London practices was ~50%, compared with ~60% in East England. Overall, uptake of influenza vaccine (~80%) was the highest among all three vaccine types, followed by pneumococcal vaccine (~70%) and shingles vaccine (~50%) (figure 1).

### Vaccination uptake

Vaccination uptake differed by ethnicity, deprivation, household size and health conditions (figure 1). In multivariable analysis compared with the white population, those from black Caribbean, black African, Chinese and Other ethnic groups showed lower uptake for all three vaccines (figure 2). Influenza vaccination uptake was significantly lower in black Caribbean (OR 0.68, 95% CI 0.64 to 0.71), black African (OR 0.72; 95% CI 0.68 to 0.77), Chinese (OR 0.49; 95% CI 0.45 to 0.53) and 'other ethnic group' (OR 0.63; 95% CI 0.60 to 0.65), but there was significantly higher uptake in Indian (OR 1.21; 95% CI 1.14 to 1.28), Pakistani (OR 1.39; 95% CI 1.28 to 1.52) and Bangladeshi (OR 2.68; 95% CI 2.38 to 3.01) ethnic groups compared with the white group.

There was a similar pattern observed for pneumococcal vaccination uptake: black Caribbean (OR 0.70; 95% CI 0.66 to 0.75), black African (OR 0.56; 95% CI 0.51 to 0.62), Chinese (OR 0.49; 95% CI 0.45 to 0.53), 'other ethnic group' (OR 0.58; 95% CI 0.55 to 0.61) and also additionally for other Asian (OR 0.87; 95% CI 0.80

**Figure 1** Box and whiskers diagrams summarising influenza, pneumococcal and shingles vaccination uptake/refusal rates in practices across different regions in England. The midline of box represents median uptake/refusal rate, lower and upper boundaries of box represent first and third quartile, lower and upper whiskers represent minimum and maximum rates. Each individual dot represents individual practice uptake/refusal rates.

to 0.93). Pneumococcal vaccine uptake was significantly higher only in the Bangladeshi ethnic group (OR 1.46; 95% CI 1.29 to 1.65) compared with the white group. For shingles vaccine uptake, there was significantly lower uptake in all ethnic minority groups except in Indians (OR 0.98; 95% CI 0.91 to 1.05).

For all three vaccines, vaccine uptake was generally lower among the more deprived, with the most deprived having 6%–33% lower odds of vaccine uptake (ORs 0.67–0.94) compared with the most affluent. People in households with two people had 22%–32% higher odds of having a vaccine compared with one-person households. However, the odds were lower in household sizes above three, with people in households of 10 or more people having 17%–63% lower odds of vaccine uptake compared with one-person households.

The uptake of each vaccination was also generally associated with increasing number of health conditions, with

asthma being associated with higher uptake of all three vaccines, while AF, CCF, dementia, and severe mental illness were associated with lower uptake of all three vaccines. Individuals with COPD, diabetes and immunosuppression were also associated with higher uptake of both influenza and pneumococcal vaccines, but not shingles vaccine (online supplemental figure S2).

### Vaccination refusals in the unvaccinated
There were consistently significantly higher odds of vaccine refusal among the black Caribbean group compared with the white group for all three vaccines; influenza (OR 1.45; 95% CI 1.34 to 1.56), pneumococcal (OR 1.29; 95% CI 1.14 to 1.46) and shingles (OR 1.35; 95% CI 1.23 to 1.49). Indian, Pakistani, Bangladeshi, other Asian, Black African, Chinese and other ethnic groups were significantly less likely to refuse all three vaccines compared with the white ethnic group, except

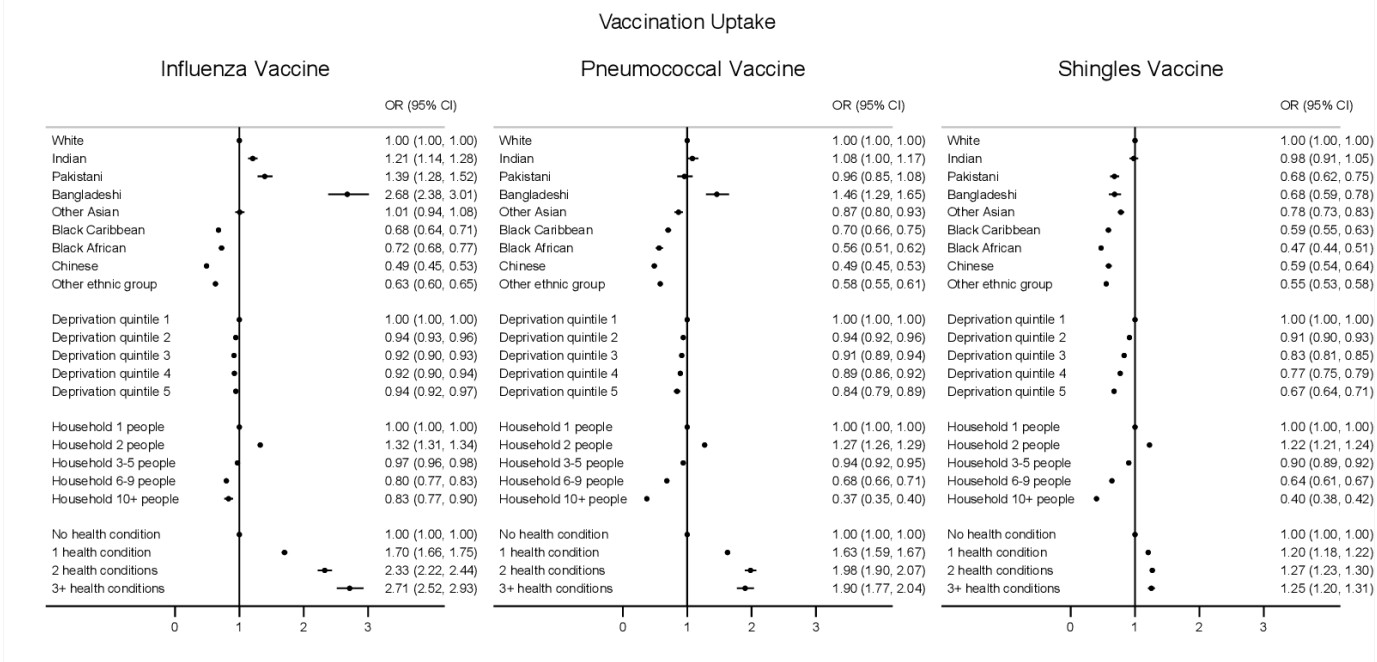

**Figure 2** Associations of ethnicity, deprivation, household size and number of health conditions on influenza, pneumococcal and shingles vaccine uptake. Logistic models for ethnicity, deprivation, household size and health conditions were run separately as each exposure factor required different sets of adjustment variables as informed by DAG evaluation. The following adjustment covariates were included in each of these models as the following: (1) Ethnicity—no adjustment; (2) Deprivation—adjusted for age, sex, region, ethnicity, household size; (3) Household size—adjusted for age, sex, region, ethnicity, deprivation, (4) Health conditions—adjusted for age, sex, region, ethnicity, deprivation, household size, house type, smoking and BMI. BMI, body mass index; DAG, directed acyclic graph.

for Pakistani and Bangladeshi, which showed no significant association with shingles vaccine refusal (figure 3).

There was a general trend of refusal with increasing deprivation, particularly with shingles vaccine in the two most deprived quintiles (OR 1.21; 95% CI 1.15 to 1.28 and OR 1.23; 95% CI 1.14 to 1.33) (4th and 5th deprivation quintiles, respectively). Higher household size was associated with lower odds of refusal of all three vaccines in households of 3+ people and more (figure 3).

In unvaccinated individuals with three or more health conditions, the odds of refusal were: influenza vaccine (OR 10.29; 95% CI 7.38 to 14.37), pneumococcal vaccine (OR 2.55; 95% CI 2.24 to 2.90), shingles vaccine (1.60; 95% CI 1.48 to 1.73). Individuals with type 2 diabetes consistently showed higher vaccine refusal for all three vaccines and individuals with COPD was also associated with higher refusal for influenza and pneumococcal vaccines (online supplemental figure S3).

**Additional analyses**
Further, we explored interactions for vaccine uptake between ethnicity and deprivation, house size and number of health conditions. First, results suggested that individuals from certain ethnic minority groups who were more deprived could be more likely to receive a vaccine, particularly Bangladeshi and Black African (online supplemental figure S4). Second, across all three vaccines evaluated, Bangladeshi individuals living in larger households could be more likely to receive a vaccine (online

supplemental figure S5). Third, vaccine uptake was generally more likely in individuals with higher number of health conditions, although the magnitude of effect varied slightly across different ethnic groups (online supplemental figure S6).

Finally, we performed sensitivity analyses to evaluate associations of vaccine uptake and refusal using complete-case analyses. In this analysis, we excluded individuals with missing information on covariates that is, ethnicity, deprivation, BMI and smoking. Results in the online supplemental figures S7 and 8 showed that estimates were comparable with the multiply imputed analysis presented as our main findings above.

**DISCUSSION**
**Summary**
In this study, we observed generally lower uptake of influenza, pneumococcal and shingles vaccinations in particular ethnic minority groups and deprived populations. Black Caribbean, black African, Chinese and other ethnic groups consistently showed lower uptake of all three vaccines studied compared with the white ethnic group. In the unvaccinated population, the black Caribbean ethnic group consistently showed increased odds of refusal for all three vaccines. More deprived populations also showed lower vaccine uptake with higher recorded refusals in the unvaccinated. Household sizes above three

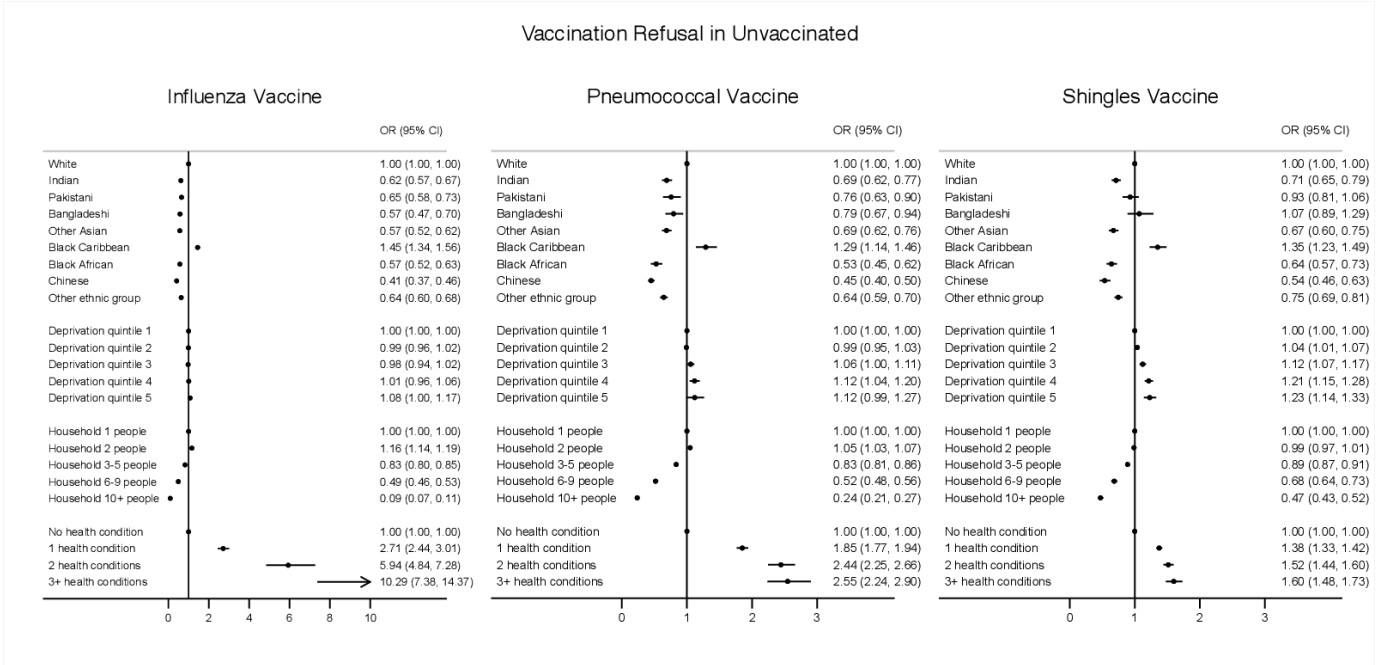

**Figure 3** Associations of ethnicity, deprivation, household size and number of health conditions on influenza, pneumococcal and shingles vaccine refusal in the unvaccinated population. Logistic models for ethnicity, deprivation, household size and health conditions were run separately as each exposure factor required different sets of adjustment variables as informed by DAG evaluation. The following adjustment covariates were included in each of these models as the following: (1) Ethnicity—no adjustment; (2) Deprivation—adjusted for age, sex, region, ethnicity, household size; (3) Household size—adjusted for age, sex, region, ethnicity, deprivation, (4) Health conditions—adjusted for age, sex, region, ethnicity, deprivation, household size, house type, smoking and BMI. BMI, body mass index; DAG, directed acyclic graph.

persons were associated with lower vaccine uptake, but were not associated with higher refusal. Further, a lower number of pre-existing health conditions was generally associated with lower odds of vaccine uptake, although this was not reflected in terms of higher odds of refusal.

### Comparison with existing literature

Our observations that influenza vaccination uptake is inversely correlated with deprivation and varies across ethnic groups build on results from a recent study of adults between 2011 and 2016 using the CPRD database.[7] This study analysed seasonal influenza vaccination uptake across five 'seasons' and similarly found that in the over 65s, black individuals were significantly less likely than white individuals to receive this vaccination. However, our study finds that South Asians may be more likely to have higher uptake of influenza vaccine, which may warrant further qualitative study to examine potential socioeconomic and behavioural factors driving this observation. Our examination of three vaccinations within a larger sample size (over 2 million vs 611 000), a more granular categorisation of ethnic groups (9 vs 4) and regions (10 vs 4), improved handling of missing data, and our analysis of vaccination refusals in the unvaccinated substantially improves our understanding of these complex public health behaviours. Our results showed that although four ethnic minority groups (black Caribbean, black African, Chinese and other ethnic group) had lower uptake of

influenza vaccine, only the black Caribbean group showed increased odds of refusal among the unvaccinated.

We also found lower vaccine uptake in household sizes above three persons, although they also showed lower refusals in the unvaccinated population. This suggests that lower vaccine uptake in larger households could be driven by barriers to vaccine uptake other than due to refusal alone. A study in Hong Kong showed that vaccine uptake in the elderly living with younger family members was lower compared with elderly individuals living alone, or living with other elderly household members.[6] This calls for further ethnographic research to explore social and household characteristics including age structure of household members and its potential association with vaccine uptake in the elderly in England.

Higher uptake of influenza and pneumococcal vaccinations in individuals with asthma, COPD, diabetes and immunosuppression could be related to clinical guidelines where individuals in these clinical risk groups would be more likely to be offered a vaccine by their healthcare providers.[22 23] On the contrary, lower vaccine uptake in those with fewer health conditions could potentially be attributable to reduced contact with health services in the healthier population and hence, reduced likelihood to receive 'opportunistic' vaccination offers. Despite that, it is worth noting that our study also found that in the unvaccinated population there remains significant

refusal in those with type 2 diabetes and COPD. Possibly relevant factors could be resistance to lifestyle and behaviour changes, in which individuals with diabetes and COPD who might be more likely to have unhealthy lifestyles, for example, smoking,[24 25] might also be less receptive to health interventions, i.e. vaccines. However, this finding needs confirmation in other studies. In addition, interaction analyses in our study showed that certain ethnic minority groups such as Bangladeshis who were more deprived and living in larger households were more likely to receive a vaccine. This could potentially be due to availability of outreach programmes organised by local communities and GPs in these areas to create awareness and provide health education.[26 27]

Vaccine hesitancy findings from this study may also be relevant to ongoing COVID-19 vaccine hesitancy in the population. In a population study in older adults using National Immunisation Management System in the England, UK, it has been similarly shown that black African and black Caribbean and more deprived populations were less likely to receive COVID-19 vaccine.[28] These similarities in findings across different vaccines suggest possible shared drivers of vaccine hesitancy, which might help inform future public health strategies for equitable implementation of vaccination programmes in general.

### Strengths and limitations

Use of the QResearch database offered a population-representative study sample with accurately coded data, enabling capture of vaccinations occurring outside GP (such as in pharmacies), as well as recorded invitations to vaccination sent by GPs and patient refusals. This permitted a robust evaluation of not only uptake, but also possible contributory mechanisms leading to uptake behaviours. Limitations include the lack of recording of variables such as religion, personal beliefs and reasons for refusal that predicate vaccine hesitancy in our sample. Further, our dataset also did not capture literacy levels, language barriers, access and education status, and hence were not able to evaluate the impact of these socioeconomic factors on vaccination uptake and refusal. These could be important factors influencing complex decision making and behavioural aspects, and hence would warrant further qualitative and ethnography studies. Classification of vaccination-related endpoints was reliant on individual practitioners using Read and SNOMED codes on the EMIS software system; however, as GP surgeries are financially incentivised through 'Quality Outcome Framework' payments to record vaccination services and we used an appropriately wide range of codes in our endpoint definitions, the risk of misclassification may be low.

### Implications for research and practice

Two key principles in health inequalities are Tudor-Hart's inverse care law,[29] where service provision is inversely proportional to the need for it, and the inverse equity hypothesis, which posits that new healthcare interventions are most likely to be taken up by those in less need and thus exacerbate pre-existing inequality in the short term. Our study may help inform policy makers regarding reducing inequity in the uptake of the studied vaccines, and tailor public health messaging to diverse communities. Elucidating the extent to which ethnic patterns in vaccine refusal are driven by cultural perceptions, institutional mistrust, variation in penetrance of misinformation and structural barriers for example, transport, language and occupational barriers in different ethnic groups requires further study in robust surveys and qualitative research. This may inform tailoring of information dissemination strategies and misinformation countermeasures to specific groups and geographical areas. Furthermore, judicious, longitudinal monitoring of the uptake and refusal rates of vaccines in different ethnic and social groups should enable real-time assessment of developing inequalities, which may inform adaptive public health strategies. Data from this may help develop strategies for increasing uptake in these groups including developing information about vaccines in different languages for use by community leaders, faith groups, local healthcare providers and community champions.[30]

### CONCLUSIONS

Certain ethnic minority, deprived populations, large households and healthier individuals were less likely to receive a vaccine, although in the unvaccinated population, higher odds of refusal were only associated with ethnicity and deprivation, but not larger households nor comorbidities. Understanding these associations may inform tailored public health messaging to different communities for equitable implementation of vaccination programmes.

**Author affiliations**
[1]Nuffield Department of Primary Care Health Sciences, University of Oxford, Oxford, UK
[2]Primary Care Research Centre, University of Southampton Faculty of Medicine, Southampton, UK
[3]University of Leicester, Leicester, UK
[4]Diabetes Research Centre, University of Leicester, Leicester, UK
[5]Sunnybrooke Health Sciences Centre, Toronto, Ontario, Canada
[6]Division of Primary Care, University of Nottingham, Nottingham, UK
[7]The Primary Care Unit, University of Cambridge, Cambridge, UK

**Contributors** JH-C and HD-M obtained funding for the study. JH-C extracted the data. PST, MP and AKC led data analysis and wrote first draft. PST, MP, AKC, HD-M, DS, TAR, CG, FZ, BRS, CC, SJG, KK and JH-C interpreted results, participated in critical revisions of manuscript and approved the final version. PST is the guarantor for the study.

**Funding** This project was funded by a grant from the Medical Research Council (MRC Grant Ref: MR/V027778/1). JH-C has received grants from the National Institute for Health Research Biomedical Research Centre, Oxford, John Fell Oxford University Press Research Fund, Cancer Research UK (grant number C5255/A18085) through the Cancer Research UK Oxford Centre, and the Oxford Wellcome Institutional Strategic Support Fund (204826/Z/16/Z) during the conduct of the study, is an unpaid director of QResearch, a not-for-profit organisation which is a partnership between the University of Oxford and EMIS Health who supply the QResearch database used for this work, and is a founder and shareholder of

ClinRisk and was its medical director until 31 May 2019; ClinRisk produces open and closed source software to implement clinical risk algorithms (outside this work) into clinical computer systems. AKC is funded by a Clinical Research Training Fellowship from Cancer Research UK (grant C2195/A31310). KK is supported by NIHR Applied Collaboration East Midlands and NIHR Leicester Biomedical Research Centre. The University of Cambridge has received salary support in respect of SJG from the NHS in the East of England through the Clinical Academic Reserve.We acknowledge the contribution of EMIS practices who contribute to QResearch®, EMIS Health, University of Nottingham and the Chancellor, Masters & Scholars of the University of Oxford for expertise in establishing, developing and supporting the QResearch database. The Hospital Episode Statistics datasets and civil registration data are used with permission from NHS Digital who retain the copyright for those data. The Hospital Episode Statistics data used in this analysis are Copyright © (2021), the Health and Social Care Information Centre and re-used with the permission of the Health and Social Care Information Centre [and the University of Oxford]; all rights reserved. This project involves data derived from patient-level information collected by the NHS, as part of the care and support of cancer patients. The data is collated, maintained and quality assured by the National Cancer Registration and Analysis Service, which is part of Public Health England (PHE). Access to the data was facilitated by the PHE Office for Data Release. The Office for National Statistics, Public Health England, and NHS Digital bear no responsibility for the analysis or interpretation of the data.The investigators acknowledge the philanthropic support of the donors to the University of Oxford's COVID-19 Research Response Fund.

**Disclaimer** The funder had no role in the study design, in the collection, analysis, or interpretation of data, in the writing of the report, or in the decision to submit the paper for publication.

**Competing interests** PST reports previous consultation with AstraZeneca and Duke-NUS outside the submitted work. KK is a Member of the Scientific Advisory Group for Emergencies (SAGE), Member of Independent SAGE, Director of the University of Leicester Centre for Black Minority Health and Trustee of the south Asian Health Foundation. JH-C is a member of several SAGE committees and chair of the risk stratification subgroup of the NERVTAG. She is an unpaid director of QResearch and founder and former medical director of ClinRisk Ltd (outside the submitted work). MP, AKC, HD-M, DS, TAR, FZ, BRS, SJG, CC, CG have no interests to declare.

**Patient and public involvement** Patients and/or the public were involved in the design, or conduct, or reporting, or dissemination plans of this research. Refer to the Methods section for further details.

**Patient consent for publication** Not applicable.

**Provenance and peer review** Not commissioned; externally peer reviewed.

**Data availability statement** Data may be obtained from a third party and are not publicly available. To guarantee the confidentiality of personal and health information, only the authors have had access to the data during the study in accordance with the relevant licence agreements. Access to QResearch data is according to the information on the QResearch website (www.qresearch.org).

**ORCID iDs**
Pui San Tan http://orcid.org/0000-0003-3359-1874
Martina Patone http://orcid.org/0000-0001-5954-1045
Ashley Kieran Clift http://orcid.org/0000-0002-0061-979X
Tom A Ranger http://orcid.org/0000-0003-3091-2337
Cesar Garriga http://orcid.org/0000-0001-7073-3611
Julia Hippisley-Cox http://orcid.org/0000-0002-2479-7283

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
