## [Reviewer comments · BMJ Open]

ARTICLE DETAILS

TITLE (PROVISIONAL)	Factors influencing influenza, pneumococcal and shingles vaccine uptake and refusal in older adults: a population-based cross-sectional study in England
AUTHORS	Tan, Pui San; Patone, Martina; Clift, Ashley; Dambha-Miller, Hajira; Saatci, Defne; Ranger, Tom; Garriga, Cesar; Zaccardi, Francesco; Shah, Baiju; Coupland, Carol; Griffin, Simon; Khunti, Kamlesh; Hippisley-Cox, Julia

VERSION 1 – REVIEW

REVIEWER	Gatwood, Justin Univ Tennessee Grant funding and consultant for multiple pharmaceutical companies manufacturing vaccines
REVIEW RETURNED	19-Jan-2022

GENERAL COMMENTS	I appreciate the authors tackling the tall task of investigating factors contributing to vaccine uptake across multiple recommended vaccines in adults. Results will undoubtedly contribute to developing approaches across practice and may even have impact beyond the UK. That said, there are a few areas in which the paper could be improved. Abstract: -I am not sure "heterogeneous" is the best word to describe uptake, and "non-vaccine uptake" is equally odd-The first sentence in design is a fragment-Second sentence of the Results begins with a numeral and is a fragment, as is the sentence following it-The second sentence of the Conclusions seems to go beyond the scope of what you assessed and you should likely not offer guesses but stay within the confines of your data Background: -This section provide a sufficient overview to justify the study.-The second sentence in paragraph 2 is a bit tough to follow and why are you reporting data for ages 71 and 78? That may simply be a readability issue based on how the sentence is structured. Please clarify.-What is the difference between "non-uptake" and "refusal?" This is an important point to clarify based on how later operationalize the outcome measures. Methods: -First sentence is a fragment-Please expand on what all is in the database for those outside the
--

	UK, particularly how the data are aggregated and what they largely contain  - Please explain Townsend deprivation or otherwise clarify what "deprived" means or reflects. This is a key variable and we need more context. -It would seem that only main effects were assessed but many of the variables could have some interactivity. I would suggest considering testing interactive effects, which you could certainly do considering your sample sizes Results:  -First paragraph: please do not fully repeat details (e.g., minorities) that are fully tabled -You make a comment on practice-level data and heterogeneity of uptake but I see no tests to corroborate that claim -Figure 1 is difficult to follow and should be converted to a table -Figure 2 needs references categories listed -It is not entirely clear whether you controlled for the full set of characteristics listed or instead ran models with only those variables listed in the figures. Please indicate your full set of covariates for all models. -Were the models focusing on comorbid conditions run separately from the other factors? This is not clear and requires justification, if so. -The sensitivity analyses are only briefly detailed and if you are going to mention them we should have more than a single sentence on what was done. -Figure S5 seems to lack context/applicability. I might recommend deleting unless it can be explained in better detail. Discussion:  -Strengths and limitations are oddly placed- lead with commentary and move this back to later in the section -You discuss correlations but no statistics were offered. Please clarify or alter the verbiage. -I found the "comparisons" subsection/paragraphs a bit superficial. You have a lot to discuss but your comparison to the literature and what your findings mean comparatively are shallow at best. Please expand. This is particularly true for the inference provided on household size. -Considering the recommendations made for certain vaccines according to particular diseases, there should be more analysis/commentary on the comorbidities. This is vital information to tailoring approaches but you only focus a bit of attention on some conditions and your statistics seemed to focus more on counts rather than individual diseases. Counts are one thing but given how certain conditions predispose adults to certain complications we should expect a deeper dive into individual (or even combinations of) the diseases you extracted. -The implications subsection was a nice contribution and a strength of this section. Global comment:  -There were significant grammatical issues throughout that need addressing, particularly the number of fragments inserted. Also, can we find a better term than "non-uptake?" It just doesn't seem to fit very well.
--	--

REVIEWER	Stoto, Michael Georgetown University
REVIEW RETURNED	25-Feb-2022

GENERAL COMMENTS	Using appropriate statistical methods, this paper demonstrates the use of an important new database to study the critically important topic of vaccine uptake and refusal, which has become even more salient during the COVID-19 pandemic. The QResearch database on which this analysis is based includes records for over 10 million patients registered with more than 1800 practices in England, and seems to be representative of the general population. The size of this database, together with the availability of demographic variables, allows the authors to derive precise estimates demonstrating the how vaccine uptake and refusal vary across the population. As a result, the analysis provides solid information on WHO refused and is not vaccinated. The QResearch database does not, however, include information on variables such as religion, personal beliefs and reasons for refusal that predicate vaccine hesitancy. In addition, the database does not capture literacy levels, language barriers, access and education status, and similar variables, so the authors were not able to evaluate the impact of these socioeconomic factors on vaccination uptake and refusal. In other words, the analysis cannot say WHY individuals are not vaccinated. During the vast scientific literature on vaccine hesitancy that existed before COVID-19 and has flourished during the pandemic is primarily focused on WHY questions. While the WHO issues are surely important to public health officials in England, the inability to answer WHY questions makes the results far less relevant to the international research community. Furthermore, given the salience of vaccine hesitancy in the pandemic, it is odd that COVID-19 is not mentioned. Adding SARS-CoV-2 vaccines to the analysis would make it far more relevant to current policy discussions and perhaps connect it better to current literature on vaccine hesitancy.
---

VERSION 1 – AUTHOR RESPONSE

Reviewer: 1

Dr. Justin Gatwood, Univ Tennessee

Comments to the Author:

I appreciate the authors tackling the tall task of investigating factors contributing to vaccine uptake across multiple recommended vaccines in adults. Results will undoubtedly contribute to developing approaches across practice and may even have impact beyond the UK. That said, there are a few areas in which the paper could be improved.

Response: Thank you.

Abstract:

-I am not sure "heterogeneous" is the best word to describe uptake, and "non-vaccine uptake" is equally odd

-The first sentence in design is a fragment

-Second sentence of the Results begins with a numeral and is a fragment, as is the sentence following it

-The second sentence of the Conclusions seems to go beyond the scope of what you assessed and you should likely not offer guesses but stay within the confines of your data

Response: Thanks for the feedback. We have now rephrased "heterogenous" to "varied across different regions and socioeconomic backgrounds." and "non-vaccine uptake" to "unvaccinated". We have also fixed the fragmented sentence in Design.

In the conclusion section, we have re-worded it to keep within the boundaries of our study's findings as the following "Certain ethnic minority, deprived populations, larger households and healthier individuals were less likely to receive a vaccine, although refusal was only associated with ethnicity and deprivation."

Background:

-This section provide a sufficient overview to justify the study.

-The second sentence in paragraph 2 is a bit tough to follow and why are you reporting data for ages 71 and 78? That may simply be a readability issue based on how the sentence is structured. Please clarify.

Response: We have used uptake and refusal of influenza, pneumococcal, and shingles vaccines as our main study outcome. The time window for the outcomes was any record of vaccine uptake and refusal during the period of 1st January 1989 until 31st October 2020 (last database update). We have now added some text to the section to explain it better.

As for shingles vaccination, the national program was launched in the UK for adults aged 70 to 79 beginning year 2013. Hence, in this study, we have included individuals who were aged 70 and above, and excluding those aged 80 and above in year 2013 as they were not eligible at the time. We have edited the text in the paper to make this clearer.

Ref: <https://www.gov.uk/government/publications/shingles-vaccination-for-adults-aged-70-or-79-years-of-age-a5-leaflet/vaccination-against-shingles-guide>

-What is the difference between "non-uptake" and "refusal?" This is an important point to clarify based on how later operationalize the outcome measures.

Response: "non-uptake" originally referred to people who did not take up a vaccine, regardless of reasons e.g. refused (or "refusal"), not offered, or other reasons. However, we acknowledge that the terms have not been clear, hence we have now changed the term "non-uptake" to "unvaccinated" throughout the paper.

Methods:

-First sentence is a fragment

Response: Corrected.

-Please expand on what all is in the database for those outside the UK, particularly how the data are aggregated and what they largely contain

Response: We have added the following text to the section to give a brief description about the database:

“QRResearch is a database with over 10 million current patients registered at more than 1800 practices in England. QRResearch is an electronic healthcare primary care database in the UK with individual patient level records for general practices (GP) using the EMIS computer record system. The database captures information from GP consultations; among which includes patient demographics, socioeconomic status, diagnoses, laboratory test results, treatments and vaccinations.”

- Please explain Townsend deprivation or otherwise clarify what "deprived" means or reflects. This is a key variable and we need more context.

Response: Townsend score is an index of deprivation commonly used in the UK to measure socioeconomic status. It uses the following characteristics to measure deprivation by postcode; proportion of (1) unemployment, (2) non-car ownership, (3) non-home ownership, and (4) household crowding – with a higher score suggests greater deprivation. In this study the scores were reported in quintiles, i.e first quintile will be least deprived while fifth quintile will be most deprived. We have added text in the methods section to explain more on this.

-It would seem that only main effects were assessed but many of the variables could have some interactivity. I would suggest considering testing interactive effects, which you could certainly do considering your sample sizes

Response: We have now added to the methods and results section the following interaction analyses for vaccine uptake between ethnicity and deprivation, household size, and number of health conditions. (Supplement Figures S3-5).

Results:

-First paragraph: please do not fully repeat details (e.g., minorities) that are fully tabled -You make a comment on practice-level data and heterogeneity of uptake but I see no tests to corroborate that claim.

Response: The minorities details have been removed from the text. Secondly, Figure 1 was intended to provide a descriptive overview of vaccine uptake rates across different regions in England – we have now taken out the term “heterogeneity” and rephrased it as a descriptive overview of uptake rates.

-Figure 1 is difficult to follow and should be converted to a table – add description of Figure to footnote

Response: Compared to a table, we found that it is much easier and more informative (median, lower/upper ranges) to display differences/spread in uptake rates across different regions in a Figure. We have now provided a more detailed description of the Figure in the header to aid interpretation. We are happy to revert to a table if the editor prefers a table.

-Figure 2 needs references categories listed

-It is not entirely clear whether you controlled for the full set of characteristics listed or instead ran models with only those variables listed in the figures. Please indicate your full set of covariates for all models.

Response: Thanks, we have now added a full description of the covariates included for each model in the footnote.

-Were the models focusing on comorbid conditions run separately from the other factors? This is not clear and requires justification, if so.

Response: Logistic models for ethnicity, deprivation, household size and health conditions were run separately as each exposure factor required different sets of adjustment variables as informed by DAG evaluation - we have now added footnotes to figures to clarify this. The conditions model comprise of number of comorbidities counts (presented as a summary measure for health conditions in Figure 2), as well as individual comorbidities (additionally presented in Supplement Figure S1).

-The sensitivity analyses are only briefly detailed and if you are going to mention them we should have more than a single sentence on what was done.

Response: We have added more details on this in the last part of results section.

-Figure S5 seems to lack context/applicability. I might recommend deleting unless it can be explained in better detail.

Response: As we have used directed acyclic graphs (DAG) to evaluate and inform adjustment covariates included in each of our models, we feel that readers might want to know how we derived our sets of covariates used for adjustment in each model. We have now added a more detailed explanation in the Figure to inform how DAGs have been used in our study to guide modelling to aid readability and interpretation by readers. However, if the editor feels otherwise we can remove the diagram and provide a short paragraph to describe it instead.

Discussion:

-Strengths and limitations are oddly placed- lead with commentary and move this back to later in the section

Response: Thanks, this has now been moved to after Comparison with existing literature.

-You discuss correlations but no statistics were offered. Please clarify or alter the verbiage.

Response: We have now rephrased terms that may suggest statistical inference i.e. heterogeneity to terms that reflect a more descriptive overview.

-I found the "comparisons" subsection/paragraphs a bit superficial. You have a lot to discuss but your comparison to the literature and what your findings mean comparatively are shallow at best. Please expand. This is particularly true for the inference provided on household size.

Response: Thanks, we have now expanded on our discussion and added a few perspectives on our findings from interaction analyses which we thought were interesting.

-Considering the recommendations made for certain vaccines according to particular diseases, there

should be more analysis/commentary on the comorbidities. This is vital information to tailoring approaches but you only focus a bit of attention on some conditions and your statistics seemed to focus more on counts rather than individual diseases. Counts are one thing but given how certain conditions predispose adults to certain complications we should expect a deeper dive into individual (or even combinations of) the diseases you extracted.

Response: Yes, we agree. We have now added more text on the findings with individual comorbidities (Figure S1-2) and added more commentary in the discussion section. Further, we have also added analysis on the interactions between ethnic groups and comorbidities. (Figure S5).

-The implications subsection was a nice contribution and a strength of this section.

Response: Thank you.

Global comment:

-There were significant grammatical issues throughout that need addressing, particularly the number of fragments inserted. Also, can we find a better term than "non-uptake?" It just doesn't seem to fit very well.

Response: Thank you, we have now fixed the fragmented sentences. We have also rephrased "non-uptake" to "unvaccinated" throughout the paper.

Reviewer: 2

Prof. Michael Stoto, Georgetown University

Comments to the Author:

Using appropriate statistical methods, this paper demonstrates the use of an important new database to study the critically important topic of vaccine uptake and refusal, which has become even more salient during the COVID-19 pandemic.

Response: Thank you.

The QResearch database on which this analysis is based includes records for over 10 million patients registered with more than 1800 practices in England, and seems to be representative of the general population. The size of this database, together with the availability of demographic variables, allows the authors to derive precise estimates demonstrating the how vaccine uptake and refusal vary across the population. As a result, the analysis provides solid information on WHO refused and is not vaccinated.

The QResearch database does not, however, include information on variables such as religion, personal beliefs and reasons for refusal that predicate vaccine hesitancy. In addition, the database does not capture literacy levels, language barriers, access and education status, and similar variables, so the authors were not able to evaluate the impact of these socioeconomic factors on vaccination uptake and refusal. In other words, the analysis cannot say WHY individuals are not vaccinated.

During the vast scientific literature on vaccine hesitancy that existed before COVID-19 and has flourished during the pandemic is primarily focused on WHY questions. While the WHO issues are surely important to public health officials in England, the inability to answer WHY questions makes the results far less relevant to the international research community.

Response: The reviewer is correct in noting the strengths of QResearch (for example the size and representativeness of the study sample) but also the limitations (for example the absence of nuanced information that might increase understanding about mechanism, in other words why vaccine uptake varies between groups). Our descriptive study informs efforts to target approaches to increase vaccine uptake and highlights the need for different types of studies to identify potentially mutable determinants of vaccine hesitancy. We have added some text to the discussion to make this clear as below:

“Limitations include the lack of recording of variables such as religion, personal beliefs and reasons for refusal that predicate vaccine hesitancy in our sample. Further, our dataset also did not capture literacy levels, language barriers, access and education status, and hence were not able to evaluate the impact of these socioeconomic factors on vaccination uptake and refusal. These could be important factors influencing the complex decision-making and behavioural aspects and hence would warrant further qualitative and ethnography studies.”

Furthermore, given the salience of vaccine hesitancy in the pandemic, it is odd that COVID-19 is not mentioned. Adding SARS-CoV-2 vaccines to the analysis would make it far more relevant to current policy discussions and perhaps connect it better to current literature on vaccine hesitancy.

Response: We agree that the descriptive epidemiology of uptake of COVID-19 vaccination is an important topic. Indeed, colleagues in our research group have been separately awarded a grant by HDR UK to study COVID-19 vaccine uptake, the protocol is available from the link below:

https://www.qresearch.org/media/1304/ox107_covid_vaccine_safety_protocol.pdf However, we do not have COVID-19 vaccine data linked to this study which started early on during the pandemic in 2020 before the availability of COVID-19 vaccines. Furthermore, we believe that uptake of the

vaccines described in this paper is an important and ongoing issue in its own right. In addition, the descriptive epidemiology of uptake of these three vaccines, which are available to all within certain age bands, may differ from the uptake of COVID vaccines for which the vaccination programme is very different.

VERSION 2 – REVIEW

REVIEWER	Stoto, Michael Georgetown University
REVIEW RETURNED	30-May-2022

GENERAL COMMENTS	I understand and appreciate the authors' response to my comments on the earlier version of this paper. However, the world is in the midst of a multi-year natural experiment in which vaccine hesitancy has been an important factor. There's also evidence, at least for childhood diseases in the United States, that some of the hesitancy towards Covid vaccines has now spilled over into other vaccines. So, while I understand the limitations of the database, the result is a study in which some of the key factors influencing vaccine hesitancy are not addressed. The additional limitations language that the authors have added helps. But a more complete approach would mention the current Covid findings to highlight just how much of a limitation this is.
---

VERSION 2 – AUTHOR RESPONSE

Reviewer: 2

Prof. Michael Stoto, Georgetown University

Comments to the Author:

I understand and appreciate the authors' response to my comments on the earlier version of this paper. However, the world is in the midst of a multi-year natural experiment in which vaccine hesitancy has been an important factor. There's also evidence, at least for childhood diseases in the United States, that some of the hesitancy towards Covid vaccines has now spilled over into other vaccines.

So, while I understand the limitations of the database, the result is a study in which some of the key factors influencing vaccine hesitancy are not addressed. The additional limitations language that the authors have added helps. But a more complete approach would mention the current Covid findings to highlight just how much of a limitation this is.

Response: The reviewer rightly highlighted the value of learning from findings of this study in vaccine hesitancy and see how it could be helpful to also addressing COVID-19 vaccine hesitancy. We have now added the text below in the discussion to discuss this –

“Vaccine hesitancy findings from this study may also be relevant to ongoing COVID-19 vaccine hesitancy in the population. In a population study in older adults using National Immunisation Management System (NIMS) in the England, UK, it has been similarly shown that ethnic minority Black African and Black Caribbean and more deprived populations were less likely to receive COVID-19 vaccine.²⁸ These similarities in findings across different vaccines suggest possible shared drivers of vaccine hesitancy, which might help inform future public health strategies for equitable implementation of vaccination programs in general.

”

VERSION 3 – REVIEW

REVIEWER	Stoto, Michael Georgetown University
REVIEW RETURNED	06-Jul-2022

GENERAL COMMENTS	I agree with the addition that the authors made in response to my previous comments about the potential relevancy of the QResearch database to the Covid pandemic. I believe, however, that they missed the main point from my first comment, that the QResearch database does not include information on variables such as religion, personal beliefs and reasons for refusal, literacy levels, language barriers, access and education status, and other variables associated with vaccine hesitancy. These are all factors brought to light in the Covid pandemic, but are presumably important for other vaccines, especially after Covid. The unavailability of such variables is a serious LIMITATION in the QResearch database that limits the value of future analyses based on it.
---